# *Yuzhoua juvenilis*: Another Angiosperm Seen in the Early Permian?

**DOI:** 10.3390/life15020286

**Published:** 2025-02-12

**Authors:** Xin Wang, Yinggang Lei, Qiang Fu

**Affiliations:** 1State Key Laboratory of Palaeobiology and Stratigraphy, Nanjing Institute of Geology and Palaeontology and CAS Center for Excellence in Life and Paleoenvironment, Chinese Academy of Sciences, Nanjing 210008, China; 2School of Materials Science and Engineering, Henan University of Science and Technology, Luoyang 471000, China; cchighkick@163.com; 3University of Chinese Academy of Sciences, Nanjing (UCASNJ), Nanjing 211135, China

**Keywords:** angiosperm, Permian, Palaeozoic, evolution, China, *Yuzhoua*

## Abstract

“How old are angiosperms” is a frequently asked but still unanswered question. Although the underlying reason includes a lack of consensus on the criterion for fossil angiosperms, limited fossil finds, and other factors, the final answer to the question apparently lies in fossils, not wrangling among different schools. The currently mainstream idea in palaeobotany is that angiosperms cannot have existed earlier than the Early Cretaceous. This 64-year-old stereotype was recently iterated again in 2017. However, another hard-to-ignore fact is that this view is challenged by increasing pre-Cretaceous fossil evidence of angiosperms as well as molecular clock estimates. Here, we report a Permian angiosperm, *Yuzhoua* gen. nov. from Henan Province, China. This fossil plant has enclosed ovules, a defining feature idiosyncratic of angiosperms. In addition, a conspicuous style is seen on the top of the ovary, underscoring its distinction from known fossil seeds in gymnosperms. The combination of the Permian (Palaeozoic) age and these two unique features of *Yuzhoua* indicates that angiosperms first appeared much earlier than widely accepted, implying a much longer history of flowering plants. The occurrence of four specimens preserved in various states and unique morphology of *Yuzhoua* are beyond the expectations of all known theories on plant evolution, shedding new light on a previously unknown aspect of plant evolution in geological history.

## 1. Introduction

Seventy years ago, Axelord argued that angiosperms may have existed in the Palaeozoic [1], and Stebbins stated that “the evolutionary line leading to the angiosperms entered a dark tunnel of ignorance at the end of the Paleozoic until the early Cretaceous” [2]. Their conclusions have two implications: (1) angiosperms first appeared in the Palaeozoic; (2) angiosperms, although present on Earth in the Palaeozoic, are missing from the fossil record of the early Mesozoic. Obviously, such a lacuna in our knowledge about angiosperms is unacceptable in botany, as it hinders us from a comprehensive understanding of plant evolution. This explains why there are so many botanists trying to decipher the origin and evolution of angiosperms and why there is an enormous amount of disagreement among botanists. Despite Axelrod and Stebbin’s profiles and expertise in botany, their judgments are not coherent with the conclusion drawn by Scott et al. [3]. After scrutinizing all fossil evidence available then, Scott et al. rejected all pre-Cretaceous records of angiosperms in 1960 [3]. Since then, no widely accepted pre-Cretaceous angiosperms have been reported. Although there has been some research that suggests the existence of pre-Cretaceous angiosperms [4,5,6,7,8,9,10,11,12,13,14] and molecular dating has suggested the occurrence of angiosperms in the Jurassic or earlier ages [15,16], mainstream palaeobotanists insist on the ossified stereotype about angiosperm history. The latest reinforcement for such an idea was given by Herendeen et al. in 2017 [17]. The debate over angiosperm origin and history appears hard to conclude, probably because of (1) a lack of well-documented fossil angiosperms from the pre-Cretaceous, and (2) a lack of consensus on identification criteria for fossil angiosperms. Some botanists appeared to mix plesiomorph with apomorph, some proposed two different criteria for fossil angiosperms at the same time, while some others proposed a criterion that they themselves refused to adopt (for more details, see the discussion) [17,18,19,20].

Actually, it is not too hard to solve the problem of identification criteria for angiosperms. Going back to 1690, when the term “angiosperm” was coined by German botanist Paul Hermann, the term designated plants that have their seeds enclosed, in contrast to gymnosperms that have their seeds exposed to the exterior. A practical criterion for fossil angiosperms in palaeobotany is that the “ovules are completely enclosed in a carpel”, according to which an Early Cretaceous angiosperm *Archaefructus liaoningensis* was identified and published in *Science* [21]. This criterion was later refined by Tomlinson and Takaso [22], who distinguished angiosperms from gymnosperms more strictly by the timing order of two developmental events: pollination and ovule enclosure. If ovules are enclosed before pollination, then the plant unexceptionally is an angiosperm [22]. Applying this criterion, various angiosperms have been identified in the Jurassic and earlier ages [11,12,13,14]. However, the records of early angiosperms are still sparse, not enough to topple the ‘no-angiosperms-until-Cretaceous’ stereotype. To figure out the truth of angiosperm origin and evolution, adopting Tomlinson and Takaso’s criterion for angiosperms [22], here we pin down the angiospermous affinity of *Yuzhoua* from the Lower Shihhotse Formation (the Lower Permian) of Henan, China. The Permian age (>294.4 Ma) of *Yuzhoua* at least doubles the length of the widely accepted history of angiosperms, which was formerly thought to be only about 130 Ma. The unique morphology and early age of *Yuzhoua* reshape our understanding of early angiosperms and update our view on angiosperm evolution and history.

## 2. Materials and Methods

The specimens that we analyzed here were collected by YL and Chaofan Hou from an outcrop of the Lower Shihhotse Formation (Lower Permian) near Hongshanzui, Mojie town, Yuzhou city, Henan Province, China (113.17634° W, 34.15086° N), which is just outside the Cathayasian Palaeobotanical Geological Park in Henan Province, China. The palaeobotany of the region has been well studied long ago [23]. The fossil plants previously reported from this region include 307 species in 111 genera of Sphenophyllales, Equisetales, Filicales, Lepidodendrales, Noeggerathiales, Peltaspermales, Pteridospermophytes, Cycadales, Dicranohpyllopsida, Ginkgophyta, Cordaitopsida, Coniferopsida, Gigantopteridales, and plants *incertae sedis* (for detailed information, see the Appendix A) [23]. This assemblage is typical of the Lower Shihhotse Formation, Lower Permian [24]. The age is the Asselian [23], and the radiometric age is 294.4–295 Ma [25].

The specimens were preserved as coalified compressions embedded in yellowish siltstones. The general morphologies of the specimens were photographed using a Nikon D300S digital camera (Nikon, Tokyo, Japan). Details of the specimens were examined and photographed using a Nikon SMZ1500 stereomicroscope equipped with a Digital Sight DS-Fi1 camera (Nikon, Tokyo, Japan). Details of better-preserved specimens were observed and photographed using a MAIA 3 TESCAN SEM (scanning electron microscope) (Tescan GmbH, Dortmund, Germany) at the Nanjing Institute of Geology and Paleontology (NIGPAS), Nanjing, China. All photographs were saved in TIFF/JPEG format and organized together for publication using Photoshop 7.0.

## 3. Results

Genus *Yuzhoua* Wang, Lei et Fu, gen. nov.

Diagnosis: Fruit asymmetrical, elongated, including a long stalk, an ovary, and a distal style. Stalk rigid, straight, gradually transitioning to ovary. Ovary in axil of an involucre. Involucre bearing two rows of long spines along its margins and bracketing the ovary. Ovary round triangular, enclosing three oval bodies of variable sizes clustered to one side of the ovary, with a layer of tissue covering its top. Style on top of ovary, slightly curved.

Remarks: The Yuzhou Flora has been well documented previously [23]. The previously documented fossils show no resemblance to the fossils documented here. Therefore, we propose a new genus, *Yuzhoua*, for these new specimens.

Three ovules of variable morphologies are enclosed in the ovary. No seed coat or similar structure is observed in all specimens, suggesting that the oval bodies in the ovaries are ovules rather than seeds (at least not mature seeds). The presence of three ovules inside an ovary confirms the occurrence of enclosed ovules in *Yuzhoua*, suggestive of angiospermous affinity. Therefore, we propose placing *Yuzhoua* in the angiosperm category despite its incredibly old age (>294.4 Ma).

Etymology: *Yuzhoua* refers to the fossil locality in Yuzhou, Henan, China.

Type species: *Yuzhoua juvenilis* gen. et sp. nov.

Type locality: Hongshanzui, Mojie Town, Yuzhou City, Henan, China (113.17634° W, 34.15086° N).

Horizon: the Lower Shihhotse Formation.

Age: the Asselian, Early Permian (>294.4 Ma ago).                                                                            

Species *Yuzhoua juvenilis* Wang, Lei et Fu, sp. nov.

               (Figure 1, Figure 2, Figure 3, Figure 4 and Appendix A)

Species diagnosis: the same as that of the genus.

Description: The fossils documented here include six coalified compressions embedded in four blocks of yellowish siltstones (Appendix A). Some of the specimens are partially covered in sediment (Figure 1a and Appendix A). Each fruit includes an elongated stalk, an ovary, and a distal style and is 23.5~40 mm long and 6.4–10 mm wide (Figure 1a, Figure 2a, Figure 3a and Appendix A). The stalk is rigid, straight, and becomes wider distally, gradually transitioning to the ovary; it is 15~23 mm long, 0.3~0.45 mm wide in its base, and up to 5 mm wide distally (Figure 1a, Figure 2a and Appendix A). The stalk furcates into an involucre and a branch, with the ovary inserted in the axil of the involucre (Figure 1a and Figure 2a,c,e). The involucre is 8.2~13 mm long, 0.4~1.8 mm thick, and bearing two rows of long parallel protective spines along its two margins, while the branch is short, bears no spines, and is 6.5~10.8 mm long and 0.9~1.6 mm thick, appressing the ovary and terminating just below the top of the ovary (Figure 1a,b, Figure 2a,c,e and Appendix A). There are up to 10 protective spines visible on each side of the ovary, with the lengths of the visible portion varying from 1 to 7 mm (Figure 1a,b, Figure 2a, Figure 3a and Figure 4b). The ovary is inverted, round, and triangular in shape, 6.8 mm long and 5.3 mm wide, with a layer of tissue covering its top and three ovules clustered to the ovary wall near the involucre (Figure 1a–c and Appendix A). Three ovules of variable forms (the central one is triangular, the upper and lower ones are distorted oval) are 2.1~4 mm long, 1.8~2.3 mm wide, oval, and appressing each other, each with an inconspicuous oval body in it (Figure 1b,c). There is a single coalified layer partially outlining the upper and lower ovules (Figure 1b,c), which are isolated from the ovary wall and covering layer of the ovary (Figure 1b,c). The covering layer of the ovary is almost uniform in thickness, 0.24~0.27 mm thick (Figure 1b,c, Figure 2c,d and Figure 4a,b). A distal style is attached on the covering layer, slightly curved, slightly tapering distally, 4.8~12 mm long, and 0.6~0.8 mm wide at the base, tapering into 0.4~0.55 mm wide distally (Figure 1a,b, Figure 2a,b, Figure 3a and Figure 4a). The cellular details of the organ may be preserved sometimes (Figure 4c). The sketch and reconstructions of the organ are shown in Figure 5a–d.

Holotype: PB205390 (Figure 1a–c, Figure 4a–c and Appendix A).

Paratypes: PB205391, 41HIV0181, 41HIV0182 (Figure 2a–f, Figure 3a–c and Appendix A).

Collectors: Yinggang Lei, Chaofan Hou.

Collecting date: 22 June 2024.

Depository: PB205390, PB205391 are deposited in the Nanjing Institute of Geology and Palaeontology, Nanjing, China; 41HIV0181, 41HIV0182 are deposited in the Henan Natural History Museum, Henan, China.

Etymology: *juvenilis* refers to the immaturity of the fossil organ.

Remarks: The general view of the organs shown in Figure 1a, Figure 2a, and Figure 3a excludes the possibility of them being a vegetative organ, leaving only one alternative for us: a reproductive organ. Among reproductive organs, there are three types: male, female, and hermaphroditic. Since both male and hermaphroditic organs require the presence of a male part, which encloses pollen grains that have more potential to be preserved in fossils than other tissues and organs, the lack of any trace of pollen grains in our light and SEM observations eliminates the possibility of *Yuzhoua* being male and hermaphroditic organs. This reasoning and inference are further strengthened by the dimensions (2.1~4 mm long, 1.8~2.3 mm wide) and inner position of the three oval bodies in Figure 1b,c. Therefore, we conclude that what we face in these specimens is a pistil.

The style of *Yuzhoua* may be otherwise interpreted as an extension of one of the branches, as it appears to be aligned with one side of the organ. However, this possibility can be easily eliminated: as shown in Figure 4a, both the ovary wall and style have parallel sides and uniform widths below and above the covering layer of the ovary, and it is clear that their widths are distinct, implying the style cannot be an extension of the ovary wall. Instead, the style is physically connected to the ovary top (Figure 1b, Figure 2a,c and Figure 3a).

Comparable to a mature fruit, *Yuzhoua juvenilis* gen. et sp. nov. appears immature as it lacks mature seeds that have a sclerenchymatic seed coat and tend to be preserved in fossils if present. The absence of seed in *Yuzhoua juvenilis* gen. et sp. nov. implies that what we see in Figure 1c are ovules (precursors of seeds). This explains why we used *juvenilis* as the specific epithet.

## 4. Discussion

Criterion for angiosperms: In 1690, the German botanist Paul Hermann (1646–1695) coined the term “angiosperms” for plants that enclose their seeds [26], in contrast to “gymnosperms” that have their seeds exposed to the exterior. Thus, “angiospermy” was the defining feature of angiosperms, at least initially so. If this criterion could be applied properly and consistently in palaeobotany, there would be or at least less controversy about the origin and evolution of angiosperms. However, there seems to be no consensus on a criterion for angiosperms in botanical circles; on one hand, those studying extant angiosperms focus on certain taxa (orders or families within angiosperms) and never worry about the scope of angiosperms [27]. They have no need to bother them about the criterion for angiosperms. On the other hand, those studying fossil angiosperms in the Cenozoic and Late Cretaceous are dealing with angiosperms that are already full-fledged and distinct from gymnosperms [28]. They have no need to worry about a criterion for angiosperms, either. The criterion for angiosperms matters only for botanists studying early angiosperms, as, theoretically, early angiosperms should demonstrate a smooth transition and show little difference from their contemporaneous gymnosperm peers. Ironically, these palaeobotanists seem to intentionally keep a tacit agreement on the criterion for angiosperms, as it is rarely declared in publications (except Sun et al. [21]). Various schools have their own unique criteria for angiosperms, especially when involved in discussions on early angiosperms and their origin [17,18,19]. For example, Sokoloff et al. emphasized pentamery, an apomorph of eudicots, as an angiosperm-specific feature [18], while Bateman preferred closed carpels and double fertilization as “a single reproductive syndrome in extant angiosperms” [19]. A fact hard to ignore is that Bateman was a member of Sokoloff et al. So, publishing these two conflicting papers in the same year raises a question for Bateman, “Which one of these two different criteria for angiosperms is your favorite?” Intriguingly, in the same paper, Herendeen et al. proposed several features as “unique angiosperm features”, whereas they set five fossil angiosperms published by themselves as exemplar fossil angiosperms [17]. It is hard to believe that their own exemplar fossil angiosperms do not have all the features that they themselves proposed as “unique angiosperm features”. Apparently, Herendeen et al. challenged the intelligence of the readers: readers would face a difficult choice about what to follow, their criterion or their examples. It appeared that Friis et al. [29] made a clear choice, “Neither” (as Friis et al. [29] ignored or refused to cite Herendeen et al.’s work even though it was frequently taken as a milestone in palaeobotany). Examining the author lists of these two papers, it is easy to find that Herendeen et al. equates to Herendeen plus Friis et al. Such an abnormal operation implies that Friis et al. are dissatisfied with their own published paper in *Nature Plants*. To avoid and reduce chaos on the criterion for fossil angiosperms, we prefer to return to the original meaning of “angiosperm” given by Hermann, which focused on “enclosed seeds”.

A more practical criterion for fossil angiosperms was proposed and applied by Sun et al., which is the following: “The unique character of angiosperms is that the ovules are completely enclosed in a carpel” [21]. Using this criterion, Sun et al. published an Early Cretaceous angiosperm, *Archaefructus liaoningensis,* in *Science* [21]. This greatly contributed to the study of early angiosperms. Interestingly, Herendeen et al. also listed “carpels enclosing one or several bitegmic ovules with two integuments” as one of the “unique angiosperm features” [17]. It is noteworthy that both groups of authors converged on “ovules enclosed” in “carpels” as an angiosperm feature.

According to the traditional theories of angiosperm evolution, carpels in apocarpous gynoecia of Magnoliales and Ranunculales were taken as basic units of a gynoecium in angiosperms [30,31,32]. Naturally, carpels have long been a focus of botanical studies in the past decades [33,34,35,36,37,38]. However, this focus of studies becomes blurry since *Amborella* forced *Magnolia* out of the center of the stage in the molecular age [39]. Magnoliales and Ranunculales are now treated as Mesangiosperms instead of basalmost angiosperms [40]. When all angiosperms are taken into consideration, it is easy to find that *bona fide* carpels are restricted into limited apocarpous groups, and most angiosperms do not have typical carpels. Whether these angiosperms have carpels is an open question. Regardless of whether all angiosperms have carpels, an essential feature of angiosperms is still that the ovules are enclosed. To avoid unnecessary controversy, the present authors emphasize that ovules of angiosperms are enclosed in and by whatever structures.

The term “angiosperm” was coined in 1690, while the first usage of the term “carpel” was in 1835. This timing order indicates that Paul Hermann did not know of the term carpel. This explains why he did not specify which structure encloses ovules in angiosperms. So, in 1690, when he coined the term “angiosperm”, he focused on the only distinction between angiosperms and gymnosperms, namely, angiospermy and gymnospermy. Apparently, although both Sun et al. [21] and Herendeen et al. [17] emphasized ovules enclosed “by carpels”, “carpel-enclosing” as a character of angiosperms was appended to the original definition of “angiosperms” after botanists focused their studies on the assumed basalmost angiosperms with apocarpous gynoecia such Magnoliales and Ranunculales. Since these groups are not regarded as basalmost angiosperms anymore [41], whether carpels are still a feature required for all angiosperms becomes an open question. The present authors tend to reject the concept of “carpel” as it may be introduce unnecessary misunderstanding on angiosperms and may be a gratuitous extra requisite for angiosperms. This treatment is at least favored by the “noncarpellate” flower of a Jurassic angiosperm, *Nanjinganthus* [42,43,44].

The above treatment is consistent with the conclusion of Tomlinson and Takaso [22]. After a survey of pollination processes in seed plants, Tomlinson and Takaso distinguished angiosperms from gymnosperms by “ovules enclosed before pollination”, and they did not mention which structure encloses ovules in angiosperms, either [22]. The drawback of this criterion is that it incorrectly places some angiosperms (for example, *Reseda*) as non-angiosperms, while its advantage is obvious: any plant meeting this criterion is unexceptionally an angiosperm. This explains why the present authors prefer to adopt this criterion for fossil angiosperms.

Actually, this criterion has long been applied in palaeobotany implicitly. In addition to the application of this criterion on *Archaefructus liaoningensis* [21], the same criterion has been applied in other studies on early angiosperms (for example, *Schmeissneria* [11,13], *Nanjinganthus* [42,43,44], and *Taiyuanostachya* [14], to name a few). *Schmeissneria* was initially recognized as an angiosperm based on an ovule impression on the inner-ovary wall and sealed ovary tip in a Chinese material, and its seeds expected inside the fruits were later confirmed by numerous German specimens [11,13]. *Nanjinganthus* has a couple of ovules enclosed inside the ovary, but shows no trace of carpels [42,43,44]. Considering its Early Jurassic age, it is hard to derive its gynoecium in a way other than the invagination of the floral axis. This explains the lack of *bona fide* carpels in *Nanjinganthus*. *Taiyuanostachya*, which is contemporaneous with *Primocycas* (Cycadales) [45], was recognized as an angiosperm because of not only its enclosed seed but also its enclosed ovule [14]. The co-occurrence of angiospermy and angio-ovuly in *Taiyuanostachya* [14] ensures its angiospermous affinity.

Alternative Interpretations: Seeds are frequently seen in the Yuzhou Flora, for example, *Acanthocarpus acerosus* Yang, *A. obovatus* Xie, *A. superjectus* Xie, *Cardiocarpus cordai* (Gein.) Gu et Zhi, *Carpolithus beaniata* (Zhang et Mo) Yang, *C. glansiformis* Mo, *C. mitriformis* Yang, *C. taxiformis* Stockm. et Math., *Dioonocarpus ovatus* Hu et Zhu, *Holcospermum linruense* Yang, *Rhabdocarpus oliveri* Kidston, *Samaropsis pachyderma* Yang, and *Zhongzhoucarpus deltatus* Yang, to name a few [23]. Considering the Permian age, it is necessary to eliminate the possibility of *Yuzhoua* being a kind of known or unknown seed. Among the above listed, *Acanthocarpus acerosus* Yang has a 5 mm long and 1.5 mm wide “sharp needle-like beak”, *A. obovatus* Xie has a 3 mm long sharp beak, and *A. lagenarius* Halle has a 5 mm long and 2.5 mm wide “blunt conical snout” [23,46]. These seeds are the only ones known with an apical projection and are thus more or less comparable to *Yuzhoua* in the Yuzhou Flora. However, these alternatives can be easily distinguished from *Yuzhoua* by their lack of elongated stalk and protective spines. We will drop these alternatives hereafter.

*Lepidocarpon* is a noteworthy lycopod that has an unexceptionally large megasporangium, which are sometimes termed as a “seed”. Its large size and lateral laminae [47] make it more or less comparable to *Yuzhoua*, but *Lepidocarpon* lacks any structures comparable to the distinctly elongated stalk and conspicuous style of *Yuzhoua*. Furthermore, *Lepidocarpon*, as a lateral part of a strobilus, is usually aggregated into a cone, unlike the sole individual *Yuzhoua*.

If the whole structure of *Yuzhoua* were a seed, then this seed would be spiny and long-stalked. Such a combination of characters is rarely seen in known seeds. Furthermore, the seed coat expected for a seed is fully missing in *Yuzhoua*. Finally, the occurrence of more than two oval bodies (Figure 1b,c) within a seed defies interpretations as it is so far unheard of for any seed yet. This makes the seed interpretation far-fetched and hard to accept. Therefore, we drop this alternative interpretation and prefer to interpret *Yuzhoua* as an ovary that has an apical style and two or three ovules enclosed inside.

Theoretical implications: The Permian age and unique morphology of *Yuzhoua* fall beyond the expectations of all known angiosperm evolution theories. Supposedly, an early angiosperm from the Permian should have fruits comparable to those in basalmost angiosperms, such as *Amborella* (according to APG [41]) or *Magnolia* (according to previously prevailing theory [30,32]). However, the carpels of Magnoliales [48] and *Amborella* [49] are all distinct from *Yuzhoua* in morphology, especially in the lack of distal style. Such a discrepancy between theoretical expectations and actual fossil evidence casts serious doubt over these perspectives on angiosperm evolution. Apparently, theories that are only based on data from extant plants have little “predictive power” on the past angiosperms, which can only be reliably learned by studying fossil angiosperms.

A previously reported fossil, *Taiyuanostachya ovuilifera,* represents the debut of angiosperms in the Permian [14]. Although it may flourish in the Permian vegetation, representatives of angiosperms in the Permian are still too sparse or poorly understood to topple the decade-old ‘no-angiosperm-until-Cretaceous’ stereotype. To confirm the factual existence of angiosperms in the Permian, fossil plants with some features unique to angiosperms are required. *Yuzhoua* appears to meet such an expectation: distal style and enclosed ovules are two features never seen in any gymnosperms but frequently seen in angiosperms. Apparently, the discovery of *Yuzhoua* strengthens the implication given by *Taiyuanostachya ovuilifera* [14], namely, that angiosperms did occur in the Permian and in a diversified and flourished mode.

*Yuzhoua* is contemporaneous with *Primocycas chinensis* (the earliest confirmed fossil record of cycads [45,50]) and *Taiyuanostachya ovuilifera* (the earliest confirmed fossil record of angiosperms [14]), suggesting that angiosperms and cycads co-occurred in the Palaeozoic, a scenario envisioned [1,51,52] and suggested by some molecular studies in which angiosperms were taken as a sister group of all gymnosperms. Such a scenario of plant evolution has been downplayed in botany. *Yuzhoua* now seems to invite botanists to rejuvenate this scenario of plant evolution. All fossil plants contemporaneous with *Yuzhoua*, including *Taiyuanostachya ovuilifera* [14] and *Primocycas chinensis* [45], belong to the well-known Cathaysian Flora of the Lower Permian, which is characterized by the *Alethopteris–Emplectopteris–Tingia–Cathaysiopteris* assemblage that was widespread in Asia during the Early Permian [53]. The Permian age of these plants has been proven by various palaeobotanists (including leading palaeobotanists such as Dr. Xingxue Li and Dr. Zhiyan Zhou [53,54]). The occurrence of *Yuzhoua* as an angiosperm in the Yuzhou Flora, together with *Taiyuanostachya* [14], appears to start solving the puzzle raised by Stebbins decades ago [2]. Although fossil records of angiosperms are still sparse, we expect that future serious digging and studies on fossil plants in the Permian will bring up more exciting finds.

Although *Taiyuanostachya* and *Yuzhoua* both belong to angiosperms, these two taxa share few common features (including angio-ovuly), implying an unexpectedly great diversity of angiosperms in the Permian as well as an earlier origin of angiosperms (if monophyly is assumed for angiosperms). Meanwhile, it is noteworthy that both *Taiyuanostachya* and *Yuzhoua* are abundant in their local floras, as more than one of their reproductive organs have been seen on a single piece of specimen of limited dimensions (Appendix A of present paper; Appendix A of Wang and Fu [14])). These observations indicate that angiosperms were not only diversified but also abundant in Permian vegetation, at least in north China. Compared to *Taiyuanostachya*, which shows little resemblance to extant angiosperms, *Yuzhoua* bears a feature that is more frequently seen in extant angiosperms: its conspicuous distal style looks like a typical style frequently seen in extant angiosperms. The occurrence of a long style in *Yuzhoua* offers the circumventive implication that (1) its ovules are fully enclosed, (2) the competition track for pollen tube viability (hitherto only seen in angiosperms) already occurred in the Permian if a distal stigma is assumed for *Yuzhoua*. Again, the latter implication is at odds with the widely accepted assumption of primitive carpels, which are usually thought of as styleless [30], as seen in *Magnolia* or *Amborella*. The doubt cast over the current theories of angiosperm evolution by this inference cannot be overstated. Apparently, the credibility and truthfulness of many botanical theories should be reviewed carefully.

The new insights on angiosperm evolution brought by *Yuzhoua* are in line with the fossil records of insects from the Permian [55,56,57]. Pollination in angiosperms is frequently related to insects. Some angiosperms even establish species-to-species coupling relationships with certain species of insects. Although this kind of ecological relationship has been well-studied in extant ecosystems, the history of this ecological tie has been a focus of controversy among scientists. Insect pollen transportation has been suggested to have existed as early as the Permian [55,56]. The earliest record of insects with a long proboscis (a feature of insect pollinators [57]) suggests that insect pollinators had occurred by the Early Permian (Kungurian, ~276 Ma) [56,57]. Although these fossil insects were thought unrelated to angiosperms [55,56,57], this thought apparently should be reviewed while keeping our new fossil records of Permian angiosperms in mind. Such reviewing should also take the latest phylogenomic study in count, which proposed a much earlier origin of angiosperms than before [16]. It appears that data from different fields are converging to the same conclusion, making it promising to draw a more accurate picture of angiosperm evolution.

Gigantopterids with typical angiosperm venation are especially abundant in the Yuzhou Flora [23]. Although they are not found physically connected with *Yuzhoua*, it would pay doing further research to elucidate the features of reproductive organs of Gigantopterids, vegetative organs of *Yuzhoua*, and contemporaneous taxa. These works will be more intriguing when the occurrence of vessel elements (an anatomical feature otherwise almost restricted to angiosperms) in Gigantopterids [58] is taken into consideration.

## 5. Conclusions

*Yuzhoua* is a new fossil plant found in the well-studied Yuzhou Flora. Although previous works have proven the Early Permian age of the flora, our study on *Yuzhoua* reveals a previously unknown feature: enclosed ovules. Enclosed ovules are a characteristic restricted to angiosperms; thus, we propose to place *Yuzhoua* in angiosperms. This will be surprising to many since angiosperms were thought to be absent in any pre-Cretaceous age. Together with previous records of early angiosperms (especially *Taiyuanostachya* from the Early Permian), fossil insects, and molecular estimates, the discovery of *Yuzhoua* pins down the occurrence of angiosperms in the Permian. Now, the Permian appears to be a new arena for angiosperm studies, as new discoveries have broken the bottleneck in research and opened a new field for palaeobotanists who have been discouraged from thinking of pre-Cretaceous angiosperms. The contemporaneous occurrences of cycads and angiosperms in the Permian (Palaeozoic) draw a new picture of plant evolution, requiring us to refresh our thinking on the evolution of angiosperms as well as of seed plants in general.

## Figures and Tables

**Figure 1 life-15-00286-f001:**
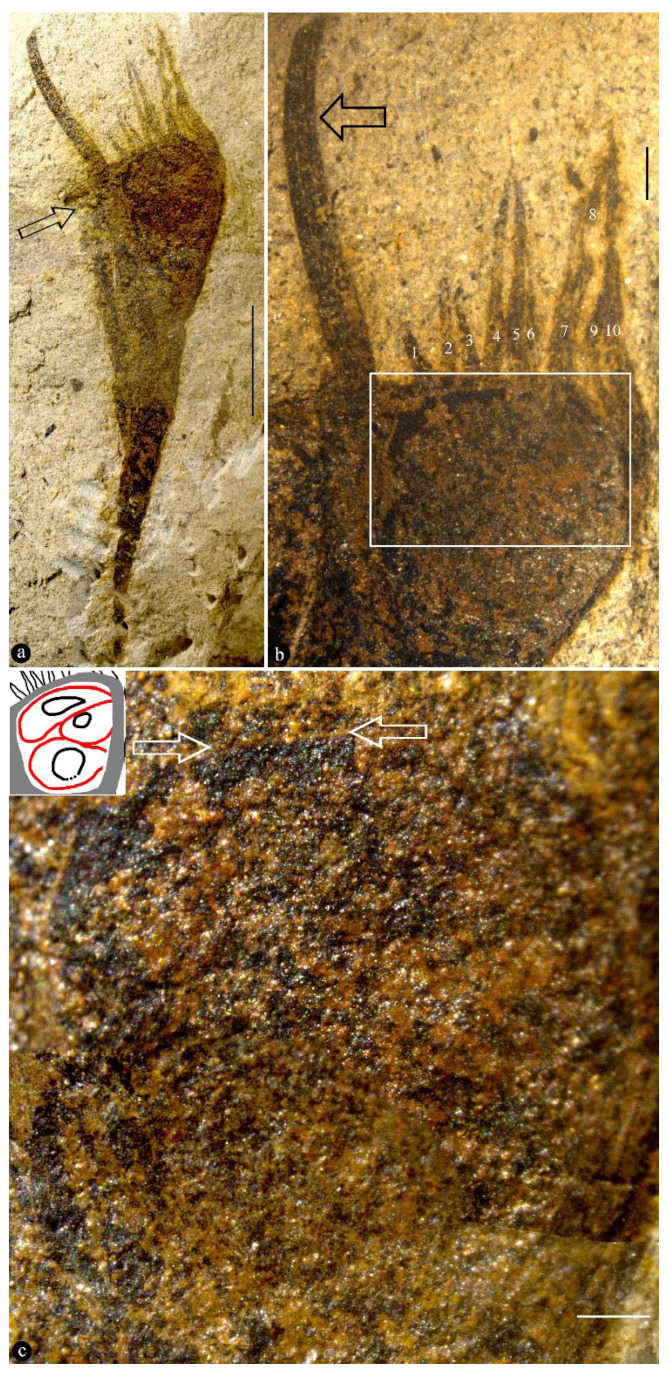
*Yuzhoua juvenilis*, general morphology and details. PB205390a. (**a**) Holotype of *Yuzhoua*. The arrow marks the tip of the branch. Scale bar = 5 mm. (**b**) Detailed view of the top portion of the ovary with a conspicuous distal style (black arrow) and several protective spins (1~10) of variable lengths. The rectangle is detailed in (**c**). Scale bar = 1 mm. (**c**) Further detailed view of the rectangle in (**b**), showing three clustered ovules inside an ovary that is secluded on the top by a layer of tissue. Note the border between the ovule and the covering layer of the ovary (between arrows). The deployment of the ovules is sketched in the upper left inset drawing. Scale bar = 0.5 mm.

**Figure 2 life-15-00286-f002:**
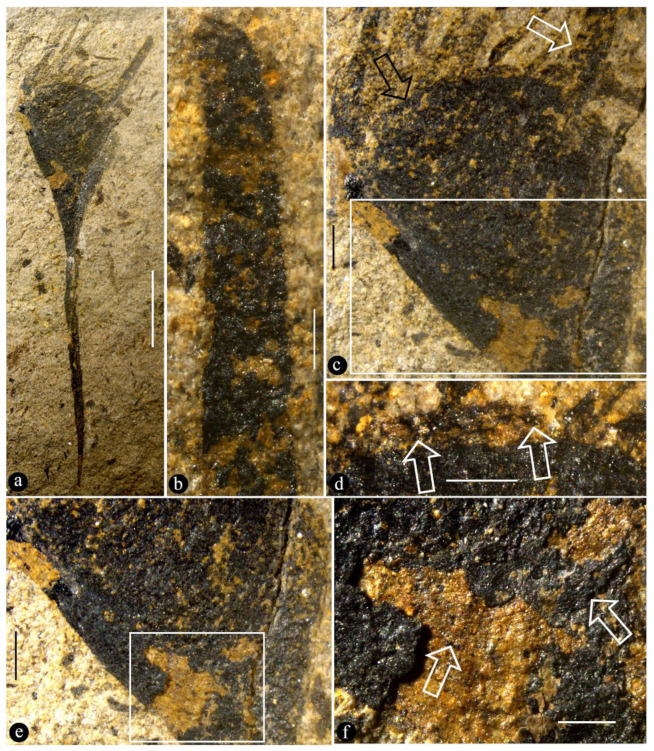
General morphology and the details of their lateral appendages. (**a**) General view. ParatPB205391. Scale bar = 5 mm. (**b**) Detailed view of the distal style. Scale bar = 0.5 mm. (**c**) Detailed view of the ovary, showing the distal style (white arrow) connected to a layer of tissue secluding the ovary (black arrow) and several protective spins. The rectangle is detailed in (**e**). Scale bar = 1 mm. (**d**) Detailed view showing the layer of tissue (white arrows) secluding the ovary. Scale bar = 1 mm. (**e**) Detailed view of the rectangle portion in (**c**). The rectangle is detailed in (**f**). Scale bar = 0.1 mm. (**f**) Detailed view of the rectangle in (**e**), showing the curving texture (arrows) of an inside ovule. Scale bar = 0.5 mm.

**Figure 3 life-15-00286-f003:**
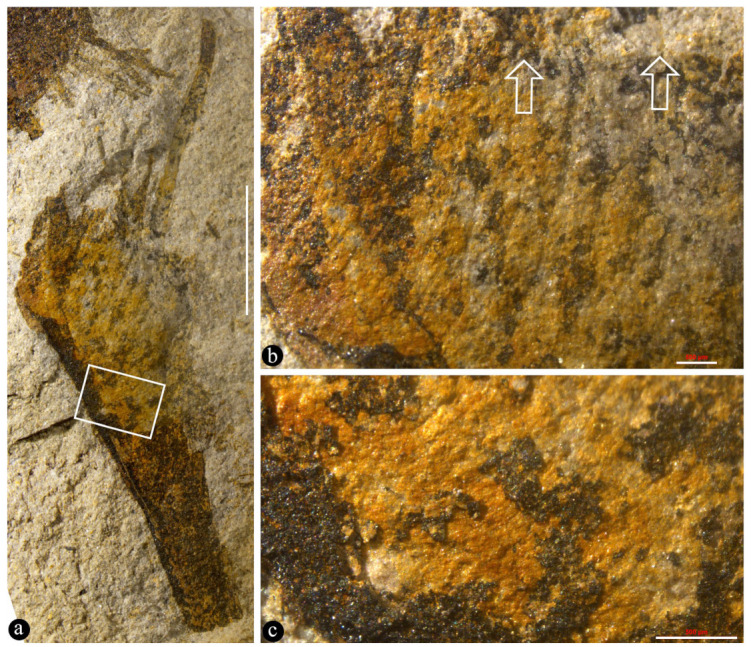
Another specimen of *Yuzhoua juvenilis* and its details. Paratype, 41HIV0181a. (**a**) General view. The rectangle is detailed in (**c**). Scale bar = 5 mm. (**b**) Detailed view of the upper portion in (**a**), showing slightly curving and parallel arrangement of the spines. Note the top of the ovary (arrows). Scale bar = 0.5 mm. (**c**) Detailed view of the rectangle in (**a**), showing curving parallel textures. Scale bar = 0.5 mm.

**Figure 4 life-15-00286-f004:**
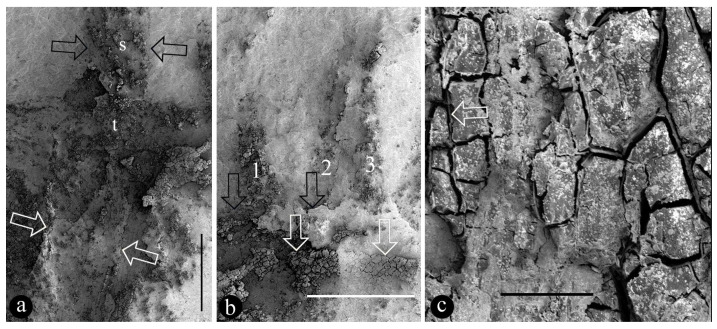
SEM details of *Yuzhoua juvenilis* shown in Figure 1a. PB205390a. (**a**) Style (s) physically connected to the top (t) of the ovary. Note the style (between black arrows) and ovary wall (between white arrows) of different thicknesses. Scale bar = 1 mm. (**b**) Detailed view of spines (1, 2, 3), top of ovary (black arrows), and ovule within the ovary (white arrows). Scale bar = 1 mm. (**c**) Cellular details of ovary wall. Note a thin cell wall (arrow) between cells. Scale bar = 0.1 mm.

**Figure 5 life-15-00286-f005:**
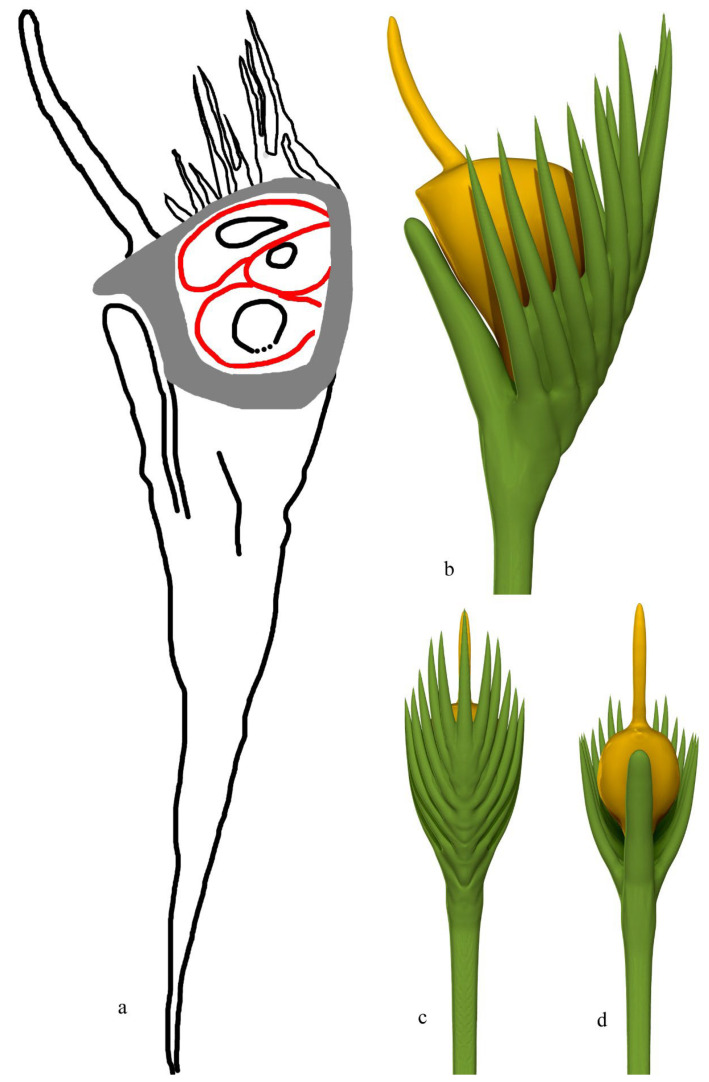
Sketch and reconstructions of *Yuzhoua* gen. et sp. nov. (**a**) Sketch of the specimen shown in Figure 1a. (**b**–**d**) Side, abaxial, and adaxial views of the organ.

## Data Availability

All data are reported in this paper.

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
