# Peer review of "Yuzhoua juvenilis: Another Angiosperm Seen in the Early Permian?"

_life, 2025, doi:10.3390/life15020286_

Round 1

Reviewer 1 Report (Previous Reviewer 2)

Comments and Suggestions for Authors

In the second version the article has been improved by the authors who better discuss the existence of a stylus in the Yuhzoua fossil. I believe the article is of great scientific interest and will stimulate the debate on the early existence of Angiosperms. It will be seen after publication whether the authors' statements will be approved by the scientific community and will stimulate further research or whether other authors will effectively try to dismantle the interpretations suggested in this paper. I believe they are more than valid arguments and I support their immediate publication. Only note, I would try to give a more effective and "visible" title to the work. In the attached file I suggest one, not too different from the current one, but just a little more effective, I think. Small typos in the attached file.

Author Response

Thanks for your help. We agree with your suggestions and comments, and have made corresponding modifications.

Reviewer 2 Report (Previous Reviewer 3)

Comments and Suggestions for Authors

This manuscript, which I have evaluated before, is, in my opinion, an important piece of scholarship and should be published. Most of the significant shortcomings that were present in earlier versions of the manuscript have now been addressed. I have only a few minor comments.

1. I recommend that the reference source (line 64) be given after "Tomlinson and Takaso's", as this source was previously mentioned quite far in the text (line 57).

2. I recommend a change in the wording of the first sentence of M&M. "Our specimens" (line 71) does not accurately describe the substance and can be interpreted in different ways because of the emphasis on ownership. This could lead to a more neutral and precise wording, for example "specimens that we have collected and analysed". 

3. I think there is no need to list the higher taxa on separate lines (lines 91-93) at the beginning of the Results. It would be sufficient to incorporate this information in one coherent sentence. However, if this suggestion was made by another reviewer, he or she must have had his or her own motives. I therefore recommend that a different way of presenting the information be considered.

4. I strongly recommend that the entire holotype label is listed in one place (line 144), as the current Code requires it. There may be no isotype label repeated as it is a duplicate of the holotype (the label information must be identical to the holotype label), only the number is sufficient. The content of the paratype labels must also be specified, not just the numbers.  This information is very important for assessing whether they are really paratypes or original material of a different rank. 

5. I recommend that references to the cited literature be added to the discussion (lines 205-218). 

Author Response

Thanks for your help. We agree with your suggestions and comments, and have made corresponding modifications according to your and the other reviewer's opinion .

This manuscript is a resubmission of an earlier submission. The following is a list of the peer review reports and author responses from that submission.

Round 1

Reviewer 1 Report

Comments and Suggestions for Authors

The authors presented an interesting hypothesis that the remains of reproductive organs found from the Permian belong to a flower plant. Unfortunately, the state of preservation of the presented fossils does not allow positive verification of this hypothesis. A number of gymnosperm species are known which had partially covered ovules. The article needs to be thoroughly rewritten. The authors did not provide sufficient evidence of the floral plant origins of the remains.

Reviewer 2 Report

Comments and Suggestions for Authors

This article is of great interest for the perspectives it opens on the origins of Angiosperms as it formulates hypotheses, in some ways revolutionary compared to current knowledge. All this derives from the accurate interpretation of Permian fossils that actually seem attributable to angiosperms. It is good to publish this interesting work and eventually other authors will refute it or accept these hypotheses of such an ancient origin of angiosperms. It could be the beginning of a revolution on this topic on which the authors have already published some contributions but now they seem more certain of their hypotheses supported by the presence of typical organs of angiosperms such as the style.

Some typos and suggestions are in the attached text

Reviewer 3 Report

Comments and Suggestions for Authors

The manuscript submitted for assessment is in line with the scope of Life magazine and presents the results of a scientifically important discovery. The paper is well prepared and there are no significant flaws. Nevertheless, I would like to make some comments that are in my opinion important.

1. I recommend replacing the title of Section 3 with a more appropriate title. This chapter presents results and the title is too general and vague (line 89). 

2. Following the provisions of the Code, and in particular Art. 38 and 46, when publishing a description of a new taxon, it is necessary to give direct credit to the authors of that taxon. Even if the authors of the newly described taxon are the same as the authors of the publication, they must be indicated. Therefore (line 90) authors of both fossil-taxa "Genus Yuzhoufructus gen. nov." (nov. is an abbreviation, so the full stop is necessary) and "Species Yuzhoufructus henanensis gen. et sp. nov." (line 113) must be indicated explicitly. In addition, it is incorrect (line 113) to specify 'gen.' to the species description, since "gen." was already indicated on line 90.

3. The type (holotype) must be specified accurately, in accordance with the Code provisions. A collection containing a type specimen or specimens is not a sufficient and explicit indication of type. It is necessary to provide the information exactly as required by the Code, so that the description of the new taxon is unambiguously linked to a specific and unambiguously identified type (holotype). 

4. The holotype should include not only the locality and the stratum but also the date of collection and the collector or collectors. 

5. Although the Code provisions do not prohibit the creation of generic names such as Yuzhoufructus, the authors should consider whether it is not possible to improve the genus name. If the genus does indeed turn out to represent the oldest angiosperm genus, the name will be written in textbooks. In languages where words are declinated or otherwise alternated according to grammatical need, such a name becomes impossible to adapt. The use of the technical term (fructus) to form the genus name is also open to criticism. 

6. The text contains a number of technical errors, in particular punctuation errors. These should be corrected by the authors. 

Round 2

Reviewer 1 Report

Comments and Suggestions for Authors

In the new version of the paper, the authors did not present any new evidence in the form of new documentation that would support their hypothesis. The organ that the authors interpret as a pistil is more like a megasporophyll.

I think the work needs to be completely rewritten. The authors should write that they found a new species of unknown affiliation.

However, classifying these fossils as flowering plants is an overinterpretation.

Author Response

[Comment 1]: In the new version of the paper, the authors did not present any new evidence in the form of new documentation that would support their hypothesis. The organ that the authors interpret as a pistil is more like a megasporophyll.

[Response 1]: Yes. There is no new evidence in the second version. The existing evidence is enough to support our interpretation (please refer to other reviewers' comments). Since our hypothesis is not changed, it is unnecessary to provide any new evidence. We appreciate your suggestion, however, if you cannot name a megasporophyll and a related reference, we cannot do any comparison you require and we cannot interpret our fossils as megasporophylls. Megasporophyll of which group? Please be specific. By the way, actualy, there is no so-called megasporphyll in plants at all, please refer to 

Miao, Y., Liu, Z. J., Wang, M. & Wang, X. Fossil and living cycads say "No more megasporophylls". Journal of Morphology and Anatomy 1, 107, (2017).

If you did not agree with the point of view in this reference, please feel free to write a paper to comment on and reject it.

[Comment 2]: I think the work needs to be completely rewritten. The authors should write that they found a new species of unknown affiliation.

[Response 2]: Please refer to the supplementary file with changes marked (in the second version of submission). You will find how many changes we have done. Yes. We have put it as a new species in a new genus. 

[Comment 3]: However, classifying these fossils as flowering plants is an overinterpretation.

[Response 3]: We understand your suspicion on our interpretation. However, our interpretation is evidenced. What is the evidence favoring your non-flowering-plant interpretation? To which group are you going to place it?

If not a flowering plant, what is your interpretation and evidence? If you could give us your reason and conclusion, we could consider and discuss them. If you could not give a single alternative, we would not have the capability to do the consideration and discussion.

Reviewer 3 Report

Comments and Suggestions for Authors

The authors have taken most of the comments into account and have revised the manuscript accordingly. I have no additional significant comments.

However, I do have one comment on the spelling of the species marker. The Classical Latin word 'iuvenis' (= young) and its derivatives are spelled 'juvenis, juvenilis' in modern Latin. Although the spelling used by the authors is not an error, most such epithets are spelled with the letter 'j' (https://www.ipni.org/). I therefore suggest that the authors use the modern spelling and to consider thoroughly the recommendations of the Code Art. 60 (https://www.iapt-taxon.org/nomen/main.php).

Author Response

[Comment 1]: The authors have taken most of the comments into account and have revised the manuscript accordingly. I have no additional significant comments.

[Response 1]: Thanks for your help and your recommendation.

[Comment 2]: However, I do have one comment on the spelling of the species marker. The Classical Latin word 'iuvenis' (= young) and its derivatives are spelled 'juvenis, juvenilis' in modern Latin. Although the spelling used by the authors is not an error, most such epithets are spelled with the letter 'j' (https://www.ipni.org/). I therefore suggest that the authors use the modern spelling and to consider thoroughly the recommendations of the Code Art. 60 (https://www.iapt-taxon.org/nomen/main.php).

[Response 1]: Thanks for your professional help and suggestion. We do have limited knowledge on the evolution of Latin. We have adopted your suggestion and changed the name accordingly.

Round 3

Reviewer 1 Report

Comments and Suggestions for Authors

As I mentioned in the last review, the authors did not present any new material that would make their speculations credible. In my opinion, the authors did not present adequate anatomical and morphological features that would indicate that the organ described is a pistil.

Author Response

[Comment]: As I mentioned in the last review, the authors did not present any new material that would make their speculations credible. In my opinion, the authors did not present adequate anatomical and morphological features that would indicate that the organ described is a pistil.

[Response]: The reviewer asked for new materials, as if more materials could increase the credibility. We do not agree with this point of view, and think this is an over-requirement. Many things have only ONE case to study in science. For example, there is only ONE universe. Could you deny the existence of universe? In our case, we have provided four specimens of the same plant and all these specimens consistently demonstrate the similar characters. This means that our interpretation is not an over-interpretation, instead that our interpretation is evidenced, grounded, and testable, and have been tested. If four is not enough, please name a specific number that is enough. Please do not forget justifying your number. Fossil evidence is not something easy to obtain. If you are a palaeobotanists, please do not over require us. Think about your situation. If you were trying to a paper with four specimens from the Antarctic or Moon, and the reviewer required you to show extra specimens, what would be your reaction? Please be realistic. We respect your right reviewing a paper and try our best to satisfy your requirement. But please do not over-require us. We are human beings, not God. We cannot satisfy ALL requirements, especially over-requirements.

We have provided more discussion on anatomy and morphology of Yuzhoua justifying why we think it is a pistil in the newer version, and hope you are satisfied now.